# Immobilization of Ir(OH)₃ Nanoparticles in Mesospaces of Al-SiO₂ Nanoparticles Assembly to Enhance Stability for Photocatalytic Water Oxidation

**Gentaro Sakamoto [1], Hiroyasu Tabe [1,2] and Yusuke Yamada [1,2,*]**

[1] Department of Applied Chemistry and Bioengineering, Graduate School of Engineering, Osaka City University, 3-3-138 Sugimoto, Sumiyoshi-ku, Osaka 558-8585, Japan; htabe@ocarina.osaka-cu.ac.jp (H.T.)

[2] Research Center for Artificial Photosynthesis, Osaka City University, 3-3-138 Sugimoto, Sumiyoshi, Osaka 558-8585, Japan

* Correspondence: ymd@a-chem.eng.osaka-cu.ac.jp; Tel.: +81-6-6605-2693

**Abstract:** Iridium hydroxide (Ir(OH)₃) nanoparticles exhibiting high catalytic activity for water oxidation were immobilized inside mesospaces of a silica-nanoparticles assembly (SiO₂NPA) to suppress catalytic deactivation due to agglomeration. The Ir(OH)₃ nanoparticles immobilized in SiO₂NPA (Ir(OH)₃/SiO₂NPA) catalyzed water oxidation by visible light irradiation of a solution containing persulfate ion ($S_2O_8^{2-}$) and tris(2,2′-bipyridine)ruthenium(II) ion ($[Ru^{II}(bpy)_3]^{2+}$) as a sacrificial electron acceptor and a photosensitizer, respectively. The yield of oxygen (O₂) based on the used amount of $S_2O_8^{2-}$ was maintained over 80% for four repetitive runs using Ir(OH)₃/SiO₂NPA prepared by the co-accumulation method, although the yield decreased for the reaction system using Ir(OH)₃/SiO₂NPA prepared by the equilibrium adsorption method or Ir(OH)₃ nanoparticles without SiO₂NPA support under the same reaction conditions. Immobilization of Ir(OH)₃ nanoparticles in $Al^{3+}$-doped SiO₂NPA (Al-SiO₂NPA) results in further enhancement of the catalytic stability with the yield of more than 95% at the fourth run of the repetitive experiments.

**Keywords:** photocatalysis; heterogeneous catalyst; catalyst support; self-assembly; mesoporous

## 1. Introduction

Utilization of solar energy attracts much attention to produce solar fuels using abundant water as an electron source for mitigation of environmental issues [1–7]. In the photocatalytic solar-fuel production, water oxidation involving four electrons and four proton transfer has been regarded as the rate-determining step. Thus, water oxidation catalysis of various metal oxides and metal complexes has been investigated in the photocatalytic water oxidation system using a photosensitizer and a sacrificial electron acceptor such as tris(2,2′-bipyridine)ruthenium(II) ion ($[Ru^{II}(bpy)_3]^{2+}$) and persulfate ion ($S_2O_8^{2-}$), respectively [8–18]. So far, iridium oxide or iridium hydroxide nanoparticles have been reported as the most promising water oxidation catalysts (Figure 1a) [19–28]. However, iridium oxide or iridium hydroxide nanoparticles can be deactivated due to agglomeration under harsh reaction conditions.

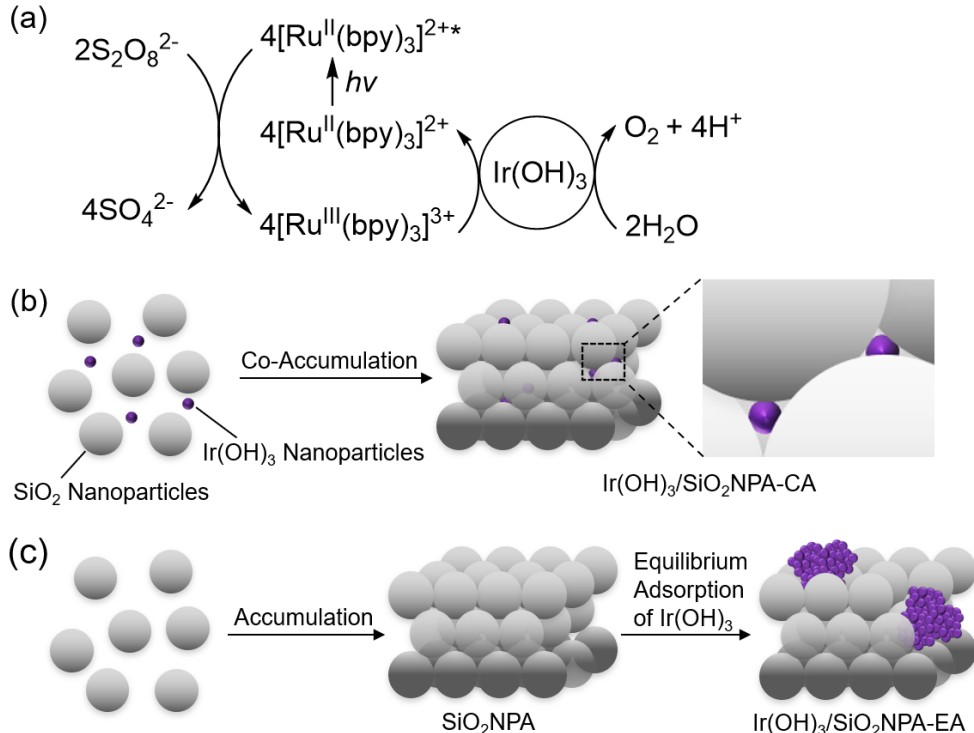

**Figure 1.** (**a**) Overall catalytic cycle of photocatalytic water oxidation under visible light irradiation using Ir(OH)$_3$ nanoparticles, S$_2$O$_8$$^{2-}$ and [Ru$^{II}$(bpy)$_3$]$^{2+}$ as a water oxidation catalyst, a sacrificial electron acceptor, and a photosensitizer, respectively. (**b**) Preparation of SiO$_2$NPA containing Ir(OH)$_3$ nanoparticles prepared via co-accumulation of SiO$_2$ and Ir(OH)$_3$ nanoparticles by drying a mixed dispersion of the nanoparticles (Ir(OH)$_3$/SiO$_2$NPA-CA). (**c**) Preparation of SiO$_2$NPA containing Ir(OH)$_3$ nanoparticles prepared by equilibrium adsorption of Ir(OH)$_3$ nanoparticles in preassembled SiO$_2$NPA (Ir(OH)$_3$/SiO$_2$NPA-EA).

Agglomeration of catalytic nanoparticles can be generally suppressed by supporting the nanoparticles on a support with high surface area [29–32]. Especially, mesoporous supports such as MCM-41 effectively suppress the agglomeration of nanoparticles because of spatial separation [33–35]. However, catalytic nanoparticles larger than the aperture size of a mesoporous support are hardly immobilized. Moreover, incorporation of catalytic nanoparticles into a mesoporous support often results in low catalytic activity because of slow diffusion of a photosensitizer to access water oxidation catalysts located inside narrow mesospaces.

Such problems can be solved by using a mesoporous support composed of a bottom-up assembly of spherical nanoparticles with uniform size [36]. The assembly possesses discrete mesospaces among nanoparticles where various molecules and nanoparticles such as organic molecules, enzymes, and metal nanoparticles are stably immobilized by encapsulation [37–41]. Recently, an assembly of aluminated silica (Al-SiO$_2$) nanoparticles in the size of ~20 nm has been reported as a support for a photocatalytic hydrogen evolution system composed of 2-phenyl-4-(1-naphthyl)quinolinium ion and platinum nanoparticles (~2 nm in diameter) as a photosensitizer and hydrogen-evolution catalysts, respectively [41]. Photocatalytic activity of the composite catalyst was enhanced compared with that using conventional mesoporous silica as a support, because the discrete mesospaces among Al-SiO$_2$ nanoparticles are suitable for electron transfer to highly dispersed platinum nanoparticles from multiple photosensitizers nearby. The structure was easily fabricated by co-accumulation of Al-SiO$_2$ and platinum nanoparticles with the photosensitizer.

We report herein the immobilization of iridium hydroxide (Ir(OH)$_3$) nanoparticles in mesospaces of an assembly of silica or aluminated silica nanoparticles (SiO$_2$NPA or Al-SiO$_2$NPA, respectively) for

the enhancement of catalytic stability during photocatalytic water oxidation. The stability of a series of water oxidation catalysts was evaluated by the repetitive experiments for photocatalytic water oxidation using [Ru$^{II}$(bpy)$_3$]$^{2+}$ and S$_2$O$_8$$^{2-}$ as a photosensitizer and a sacrificial electron acceptor, respectively. Ir(OH)$_3$/SiO$_2$NPAs prepared by two different methods were compared in terms of catalytic activity and stability for the photocatalytic water oxidation. The first catalyst was prepared by co-accumulation (CA) method, in which a dispersion containing both Ir(OH)$_3$ and SiO$_2$ nanoparticles was used for co-assembly formation (Figure 1b). The second one was prepared by equilibrium adsorption (EA) method, in which Ir(OH)$_3$ nanoparticles were immobilized in preassembled SiO$_2$NPA (Figure 1c). Size effect of SiO$_2$ nanoparticles in SiO$_2$NPAs was also scrutinized on the catalysis stability during photocatalytic water oxidation. Then, the effect of surface charge modification by using Al-SiO$_2$ nanoparticles was investigated on the catalytic stability.

## 2. Results and Discussion

### 2.1. Preparation and Water Oxidation Catalysis of Ir(OH)$_3$ Nanoparticles with and without SiO$_2$NPA

Ir(OH)$_3$ nanoparticles were synthesized by a reported procedure with slight modifications [42]. Hydrogen hexachloroiridate(IV) was treated with an aqueous solution of sodium hydroxide to form Ir(OH)$_3$ nanoparticles at a raised temperature. The nature of Ir(OH)$_3$ nanoparticles prone to agglomeration was evidenced by dynamic light scattering (DLS) and transmission electron microscopy (TEM). DLS measurements indicated that the size of Ir(OH)$_3$ nanoparticles is ~40 nm (Figure S1). However, the TEM image shown in Figure 2a suggests that the size of primary particles was several nanometers in diameter, indicating that the particles size of ~40 nm observed in DLS measurements was that of secondary particles formed by agglomeration even in dilute dispersions.

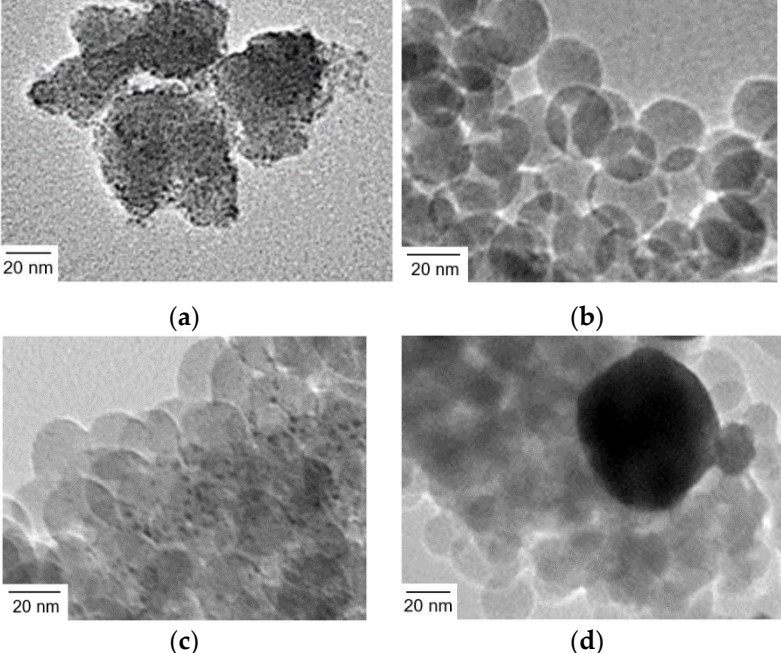

**Figure 2.** TEM images of (**a**) Ir(OH)$_3$ nanoparticles, (**b**) SiO$_2$ nanoparticles (20 nm in diameter) assembly (SiO$_2$NPA(20)), (**c**) Ir(OH)$_3$ nanoparticles immobilized in SiO$_2$NPA(20) via the co-accumulation method (Ir(OH)$_3$/SiO$_2$NPA(20)-CA), and (**d**) Ir(OH)$_3$ nanoparticles immobilized in SiO$_2$NPA(20) via the equilibrium-adsorption method (Ir(OH)$_3$/SiO$_2$NPA(20)-EA).

The Ir(OH)$_3$ nanoparticles were immobilized in SiO$_2$NPA (20 nm in diameter, SiO$_2$NPA(20)) by co-accumulation and equilibrium adsorption methods (Ir(OH)$_3$/SiO$_2$NPA(20)-CA and Ir(OH)$_3$/SiO$_2$NPA(20)-EA, respectively). TEM measurements indicated that the primary particles of

Ir(OH)$_3$ were well dispersed in Ir(OH)$_3$/SiO$_2$NPA(20)-CA (Figure 2b,c); on the other hand, Ir(OH)$_3$ nanoparticles agglomerated in Ir(OH)$_3$/SiO$_2$NPA(20)-EA (Figure 2d). The high dispersion of Ir(OH)$_3$ nanoparticles in Ir(OH)$_3$/SiO$_2$NPA(20)-CA implies attractive interaction between Ir(OH)$_3$ and SiO$_2$ nanoparticles, although both of which possess negatively charged surfaces under the preparation conditions as evidenced by Zeta potential measurements (Figure S2). On the other hand, the Ir(OH)$_3$ agglomerates in Ir(OH)$_3$/SiO$_2$NPA(20)-EA were as large as 100 nm, suggesting that Ir(OH)$_3$ nanoparticles were preferably immobilized on the surfaces not in the mesospaces of SiO$_2$NPA(20), where the stronger electrostatic repulsion is expected in the confined mesospaces.

The amounts of Ir(OH)$_3$ nanoparticles immobilized in Ir(OH)$_3$/SiO$_2$NPA(20)-CA and Ir(OH)$_3$/SiO$_2$NPA(20)-EA were quantified by X-ray fluorescence (XRF) spectroscopy (Table 1). The molar ratio of [Ir]/[Si] in Ir(OH)$_3$/SiO$_2$NPA(20)-CA (0.072 wt%) was less than half that of Ir(OH)$_3$/SiO$_2$NPA(20)-EA (0.203 wt%). The electrostatic repulsion weakening interaction between negatively charged Ir(OH)$_3$ and SiO$_2$ nanoparticles assisted the partial leaching of Ir(OH)$_3$ nanoparticles from Ir(OH)$_3$/SiO$_2$NPA(20)-CA during the washing process. On the other hand, the large Ir(OH)$_3$ agglomerates formed on the surfaces of Ir(OH)$_3$/SiO$_2$NPA(20)-EA hardly leached during the washing process, resulting in increase of the molar ratio of [Ir]/[Si] in Ir(OH)$_3$/SiO$_2$NPA(20)-EA.

**Table 1.** Molar ratios of iridium (Ir) and silicon (Si) of catalysts before and after the photocatalytic water oxidation determined by XRF analyses.

| Catalyst | [Ir]/[Si] | | [Ir$_{after}$]/[Ir$_{fresh}$] (%) |
| --- | --- | --- | --- |
| | Fresh | After the Reaction | |
| Ir(OH)$_3$/SiO$_2$NPA(20)-CA | 0.072 | 0.019 | 26 |
| Ir(OH)$_3$/SiO$_2$NPA(20)-EA | 0.203 | 0.011 | 5.4 |
| Ir(OH)$_3$/SiO$_2$NPA(10)-CA | 0.087 | 0.032 | 37 |
| Ir(OH)$_3$/SiO$_2$NPA(50)-CA | 0.084 | 0.042 | 50 |
| Ir(OH)$_3$/SiO$_2$NPA(100)-CA | 0.109 | 0.032 | 31 |
| Ir(OH)$_3$/Al-SiO$_2$NPA(10)-CA ([Al] = 5 wt%) | 0.082 | 0.038 | 46 |
| Ir(OH)$_3$/Al-SiO$_2$NPA(20)-CA ([Al] = 1 wt%) | 0.097 | 0.035 | 36 |
| Ir(OH)$_3$/Al-SiO$_2$NPA(20)-CA ([Al] = 5 wt%) | 0.125 | 0.076 | 61 |
| Ir(OH)$_3$/Al-SiO$_2$NPA(20)-CA ([Al] = 10 wt%) | 0.077 | 0.041 | 53 |
| Ir(OH)$_3$/Al$_2$O$_3$NPA-CA | 0.064 | 0.008 | 13 |

Mesoporous structures of SiO$_2$NPA(20), Ir(OH)$_3$/SiO$_2$NPA(20)-CA, and Ir(OH)$_3$/SiO$_2$NPA(20)-EA were evidenced by nitrogen adsorption–desorption isotherms with distinct hysteresis loops classified to type IV (Figure S3). The size of mesopores (*R*) of all the samples narrowly distributed around 2.4 nm as determined by the Barrett–Joyner–Halenda (BJH) method although the pore shapes were not tubular as assumed in the theory (Table 2). The Brunauer−Emmett−Teller (BET) surface areas (*S*) of SiO$_2$NPA, Ir(OH)$_3$/SiO$_2$NPA(20)-CA, and Ir(OH)$_3$/SiO$_2$NPA(20)-EA were 109, 106, and 101 m$^2$ g$^{-1}$, respectively. Internal surface areas calculated by *t*-plots of Ir(OH)$_3$/SiO$_2$NPA(20)-CA (100 m$^2$ g$^{-1}$) and Ir(OH)$_3$/SiO$_2$NPA(20)-EA (85 m$^2$ g$^{-1}$) smaller than that of SiO$_2$NPA(20) (108 m$^2$ g$^{-1}$) resulted from partial filling of the mesospaces with Ir(OH)$_3$ nanoparticles.

**Table 2.** Brunauer−Emmett−Teller (BET) surface areas ($S$), pore size determined by the Barrett–Joyner–Halenda (BJH) plot ($R$), and inner surface areas obtained by $t$-plots ($S_{int}$) of a series of catalysts calculated from the nitrogen adsorption−desorption isotherms. [a]

| Catalyst | $S$/m$^2$ g$^{-1}$ | $R$/nm | $S_{int}$/m$^2$ g$^{-1}$ |
|---|---|---|---|
| SiO$_2$NPA(10) | 189 | 1.6 | 165 |
| SiO$_2$NPA(20) | 109 | 2.4 | 108 |
| SiO$_2$NPA(50) | 65 | 7.0 | 14 |
| SiO$_2$NPA(100) | 38 | 14 | 5 |
| Ir(OH)$_3$/SiO$_2$NPA(20)-CA | 106 | 2.4 | 100 |
| Ir(OH)$_3$/SiO$_2$NPA(20)-EA | 101 | 2.4 | 85 |
| Ir(OH)$_3$/SiO$_2$NPA(10)-CA | 185 | 1.6 | 164 |
| Ir(OH)$_3$/SiO$_2$NPA(50)-CA | 69 | 7.0 | – |
| Ir(OH)$_3$/SiO$_2$NPA(100)-CA | 37 | 16 | 4 |
| Ir(OH)$_3$/Al-SiO$_2$NPA(20)-CA ([Al] = 1 wt%) | 110 | 2.4 | 106 |
| Ir(OH)$_3$/Al-SiO$_2$NPA(20)-CA ([Al] = 5 wt%) | 116 | 4.0 | 112 |
| Ir(OH)$_3$/Al-SiO$_2$NPA(20)-CA ([Al] = 10 wt%) | 95 | 7.0 | – |
| Al$_2$O$_3$NPA(20)-CA | 96 | 12 | – |
| Ir(OH)$_3$/Al$_2$O$_3$NPA(20)-CA | 96 | 6.0 | – |

[a] Nitrogen adsorption−desorption isotherms are shown in Figure S3.

Photocatalytic water oxidation was examined under visible light irradiation of a phosphate buffer solution (50 mM, 2.0 mL, pH 8.0) containing a catalyst (5.0 mg, [Ir] = 0.072 mM), tris(2,2′-bipyridine)ruthenium(II) sulfate (Ru$^{II}$(bpy)$_3$SO$_4$, 1.0 mM) as a photosensitizer and sodium persulfate (Na$_2$S$_2$O$_8$, 5.0 mM) as a sacrificial electron acceptor, respectively, by a white LED lamp ($\lambda$ > 400 nm, 33.2 mW cm$^{-2}$) at room temperature. The catalytic activity of Ir(OH)$_3$ nanoparticles without a support gradually decreased in the repetitive runs (Figure 3). The O$_2$ yield of 78% at the 1st run based on the used amount of Na$_2$S$_2$O$_8$ gradually decreased to 29 % at the 4th run. Ir(OH)$_3$/SiO$_2$NPA(20)-EA showed high catalytic activity (>85% based on the used amount of Na$_2$S$_2$O$_8$) up to the 2nd run compared with Ir(OH)$_3$ nanoparticles without a support; however, the O$_2$ yield was as low as 61% and 37% at the 3rd and 4th runs, respectively. High O$_2$ yield (>80%) even at the 4th run for the reaction system using Ir(OH)$_3$/SiO$_2$NPA(20)-CA as the catalyst indicated that the CA method is a suitable preparation method for the improvement of catalytic stability compared with the EA method.

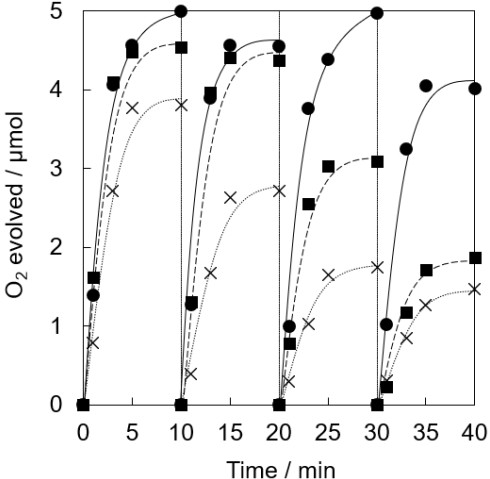

**Figure 3.** Time courses of O$_2$ evolution by visible light irradiation (white light) of a phosphate buffer solution (50 mM, 2.0 mL, pH 8.0) containing Ir(OH)$_3$/SiO$_2$NPA(20)-CA (5.0 mg, ●, solid lines), Ir(OH)$_3$/Al-SiO$_2$NPA(20)-EA (5.0 mg, ■, dashed lines) or Ir(OH)$_3$ (0.072 mM, ×, dotted lines) in the presence of [Ru$^{II}$(bpy)$_3$]SO$_4$ (1.0 mM) and Na$_2$S$_2$O$_8$ (5.0 mM) in four repetitive experiments.

XRF measurements of Ir(OH)$_3$/SiO$_2$NPA(20)-EA after the reactions clearly indicated that leaching of Ir(OH)$_3$ nanoparticles resulted in the lower durability (Table 1). The molar ratio of [Ir]/[Si] of the catalytic samples after the reactions decreased to 0.011% from 0.203%, suggesting that most Ir(OH)$_3$ nanoparticles in Ir(OH)$_3$/SiO$_2$NPA(20)-EA (95%) was lost after the reaction. The leaching of Ir(OH)$_3$ nanoparticles was suppressed in Ir(OH)$_3$/SiO$_2$NPA(20)-CA, in which more than 25% of Ir(OH)$_3$ nanoparticles remained in SiO$_2$NPA(20), because Ir(OH)$_3$ nanoparticles were preferably immobilized inside mesospaces as evidenced by TEM images as well as higher internal surface areas without agglomeration. DLS measurements of an aqueous dispersion of Ir(OH)$_3$/SiO$_2$NPA(20)-CA stirring vigorously were performed every 10 min to confirm the deformation of assembled structure of SiO$_2$NPA(20) by granulation (Figure 4). Assembly of Ir(OH)$_3$/SiO$_2$NPA(20)-CA in the size of 60–100 nm found after stirring for 30–40 min can be assigned to the secondary particles formed by deformation of the higher-order particles (300–400 nm) found in 10–20 min stirring. The deformation of SiO$_2$NPA(20)s may be attributed to the hydrolysis of siloxane bonds among SiO$_2$ nanoparticles, which can be accelerated by water diffusion in the mesospaces.

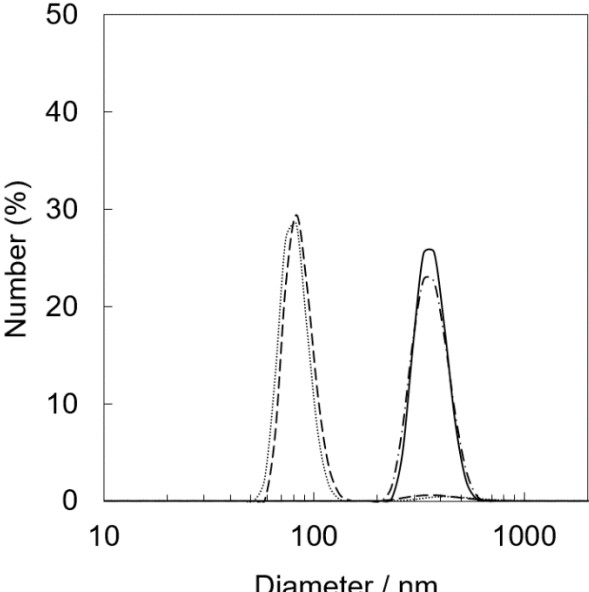

**Figure 4.** Particle size distribution of Ir(OH)$_3$/SiO$_2$NPA(20)-CA dispersed in water obtained by DLS. DLS measurements were performed after magnetically stirring (400 rpm) the dispersion for 10 min (solid line), 20 min (dashed-dotted line), 30 min (dashed line), and 40 min (dotted line).

## 2.2. Size Effect of SiO$_2$ Nanoparticles in Ir(OH)$_3$/SiO$_2$NPA-CAs on Catalytic Stability for Water Oxidation Catalysis

The size effect of SiO$_2$ nanoparticles in Ir(OH)$_3$/SiO$_2$NPA-CAs was investigated on catalytic activity and stability. SiO$_2$ nanoparticles in the size of 10, 20, 50, or 100 nm were used for the preparation of Ir(OH)$_3$/SiO$_2$NPA-CAs. Immobilization of Ir(OH)$_3$ nanoparticles was evidenced by XRF spectroscopy (Table 1). TEM observation of Ir(OH)$_3$/SiO$_2$NPA(10)-CA and Ir(OH)$_3$/SiO$_2$NPA(50)-CA indicated that Ir(OH)$_3$ nanoparticles were highly dispersed in SiO$_2$NPAs as well as Ir(OH)$_3$/SiO$_2$NPA(20)-CA (Figure 5a,b). On the other hand, some agglomerations of Ir(OH)$_3$ nanoparticles were observed on the surface of Ir(OH)$_3$/SiO$_2$NPA(100)-CA, suggesting that the larger pores formed in SiO$_2$NPA(100)-CA may not be suitable for the immobilization of primary particles of Ir(OH)$_3$ (Figure 5c).

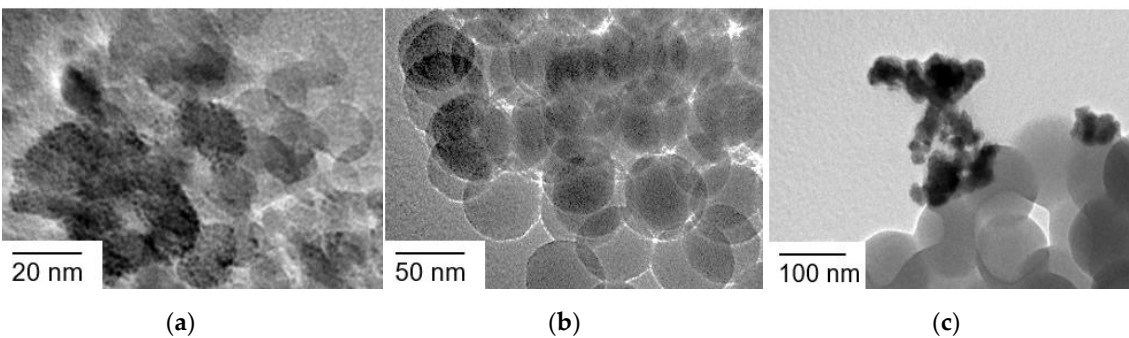

**Figure 5.** TEM images of (**a**) Ir(OH)$_3$/SiO$_2$NPA(10)-CA, (**b**) Ir(OH)$_3$/SiO$_2$NPA(50)-CA, and (**c**) Ir(OH)$_3$/SiO$_2$NPA(100)-CA.

Catalytic activity and stability of a series of Ir(OH)$_3$/SiO$_2$NPA-CAs with various sized SiO$_2$ nanoparticle for the photocatalytic water oxidation was examined in repetitive experiments. Nearly stoichiometric amount of O$_2$ evolution was evolved for Ir(OH)$_3$/SiO$_2$NPA(10)-CA and Ir(OH)$_3$/SiO$_2$NPA(20)-CA at the 1st runs (Figure 6). The O$_2$ yields decreased to 80 and 70% at the 4th runs for Ir(OH)$_3$/SiO$_2$NPA(10)-CA and Ir(OH)$_3$/SiO$_2$NPA(20)-CA, respectively. On the other hand, the O$_2$ yields at the 1st run decreased to 87 and 78% for Ir(OH)$_3$/SiO$_2$NPA(50)-CA and Ir(OH)$_3$/SiO$_2$NPA(100)-CA, respectively. The O$_2$ yields maintained higher than 90% at the 4th run for Ir(OH)$_3$/SiO$_2$NPA(50)-CA and Ir(OH)$_3$/SiO$_2$NPA(100)-CA. The comparison of Ir contents of a series of Ir(OH)$_3$/SiO$_2$NPAs-CAs in fresh and after the reaction indicated that Ir(OH)$_3$/SiO$_2$NPA(50)-CA maintained larger amount of Ir(OH)$_3$ nanoparticles after the reaction; however, half of that was leached. Thus, optimization of not only the size of mesospaces, but also surface properties of SiO$_2$NPA, is necessary for improving the catalytic stability.

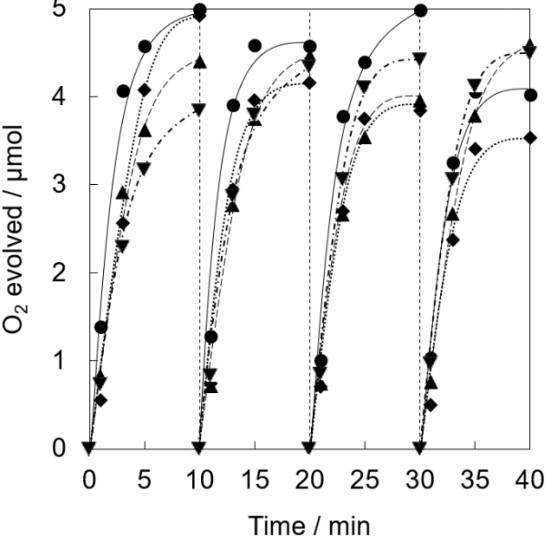

**Figure 6.** Time courses of O$_2$ evolution by visible light irradiation (white light) of a phosphate buffer solution (50 mM, 2.0 mL, pH 8.0) containing Ir(OH)$_3$/SiO$_2$NPA(10)-CA (5.0 mg, ♦, dotted lines), Ir(OH)$_3$/SiO$_2$NPA(20)-CA (5.0 mg, ●, solid lines), Ir(OH)$_3$/SiO$_2$NPA(50)-CA (5.0 mg, ▲, dashed lines), or Ir(OH)$_3$/SiO$_2$NPA(100)-CA (5.0 mg, ▼, dashed dotted lines) in the presence of [Ru$^{II}$(bpy)$_3$]SO$_4$ (1.0 mM) and Na$_2$S$_2$O$_8$ (5.0 mM) in four repetitive experiments.

### 2.3. Suppression of Leaching of Ir(OH)$_3$ Nanoparticles from SiO$_2$NPA by Surface Modification with Al$^{3+}$

Surface modification of SiO$_2$NPA with Al$^{3+}$ was examined to suppress the leaching of Ir(OH)$_3$ by lessening electrostatic repulsion. Al$^{3+}$-modified SiO$_2$ nanoparticles were synthesized by the

surface alumination of $SiO_2$ nanoparticles (20 nm in diameter) with sodium aluminate in an aqueous dispersion. $Ir(OH)_3$ nanoparticles were immobilized in an $Al^{3+}$-modified $SiO_2$ nanoparticles assembly (Al-$SiO_2$NPA) via the co-accumulation method to obtain $Ir(OH)_3$/Al-$SiO_2$NPA(20)-CA. TEM observations of $Ir(OH)_3$/Al-$SiO_2$NPA(20)-CA clearly indicated the immobilization of $Ir(OH)_3$ nanoparticles as indicated in Figure 7a (white arrows). Moreover, elemental mapping images obtained by an X-ray energy dispersive spectrometer indicated that $Al^{3+}$ ions added on $SiO_2$ nanoparticles were highly dispersed through the entire body of $Ir(OH)_3$/Al-$SiO_2$NPA(20)-CA, although the signal from Ir species was too weak to be detected (Figure 7b–d). The amount of $Ir(OH)_3$ immobilized in $Ir(OH)_3$/Al-$SiO_2$NPA(20)-CA was comparable to that in $Ir(OH)_3$/$SiO_2$NPA(20)-CA as determined by XRF measurements (Table 1). The sizes ($R$), surface areas ($S$), and inner surface areas ($S_{int}$) of the mesopores of $Ir(OH)_3$/Al-$SiO_2$NPA(20)-CA were also almost comparable to those of $Ir(OH)_3$/$SiO_2$NPA(20)-CA (Table 2).

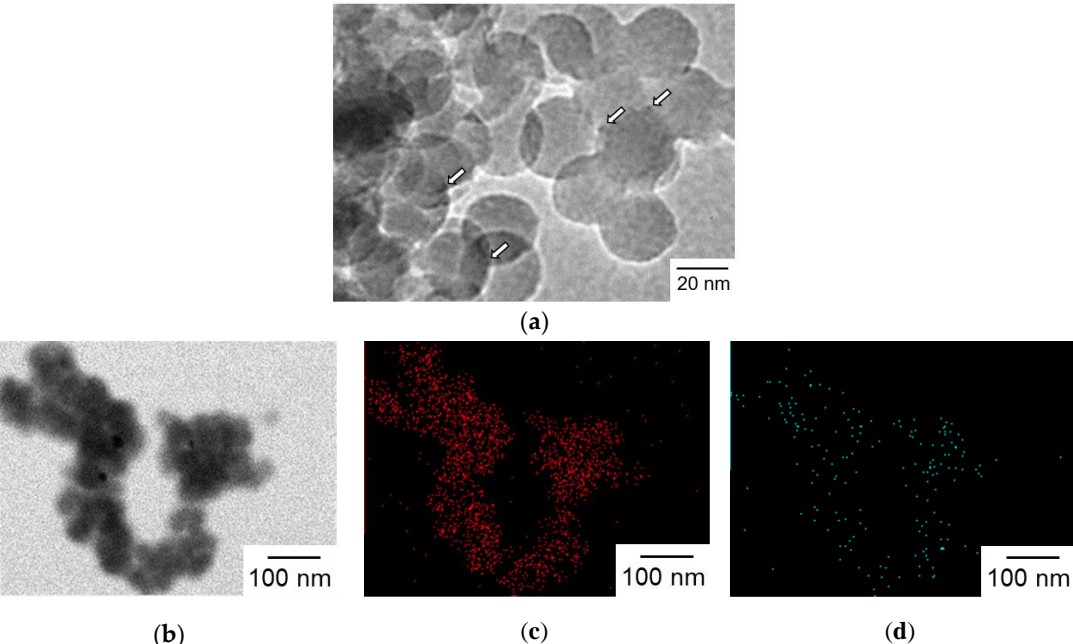

**Figure 7.** (**a**) TEM image of $Ir(OH)_3$/Al-$SiO_2$NPA(20)-CA. $Ir(OH)_3$ nanoparticles are indicated by arrows. (**b**–**d**) TEM image and elemental mappings for (**c**) silicon and (**d**) aluminum.

Photocatalytic water oxidation was carried out under visible light irradiation of a phosphate buffer solution (50 mM, 2.0 mL, pH 8.0) containing $Ir(OH)_3$/Al-$SiO_2$NPA(20)-CA (5.0 mg), $Ru^{II}(bpy)_3SO_4$ (1.0 mM) and $Na_2S_2O_8$ (5.0 mM). Nearly stoichiometric amount of $O_2$ evolution was observed for $Ir(OH)_3$/Al-$SiO_2$NPA(20)-CA (Figure 8). High catalytic activity assured by high $O_2$ yield (>93%) was maintained for four repetitive experiments. The content of Ir remained in $Ir(OH)_3$/Al-$SiO_2$NPA(20)-CA after the reaction was 61%, which is more than double that of $Ir(OH)_3$/$SiO_2$NPA(20)-CA (Table 1). On the other hand, $Ir(OH)_3$ nanoparticles immobilized in mesospaces of an alumina nanoparticles assembly ($Ir(OH)_3$/$Al_2O_3$NPA(20)-CA) showed low catalytic activity, in which lower $O_2$ yields than 60% and larger leaching amount of Ir ($[Ir_{after}]/[Ir_{fresh}]$ = 13%) in the four repetitive runs suggested that the surface modification of $SiO_2$ nanoparticles with $Al^{3+}$ is beneficial for suppressing the leaching of $Ir(OH)_3$ nanoparticles. A nearly stoichiometric amount of $O_2$ evolution was also observed for the reaction system using $Ir(OH)_3$/Al-$SiO_2$NPA(10)-CA at the 1st run (Figure 9). However, the $O_2$ yield decreased at the successive runs (50% at the fourth run) although the remaining amount of $Ir(OH)_3$ was as high as 46% (Table 1). The decreased activity could result from the lower stability of $SiO_2$NPA(10).

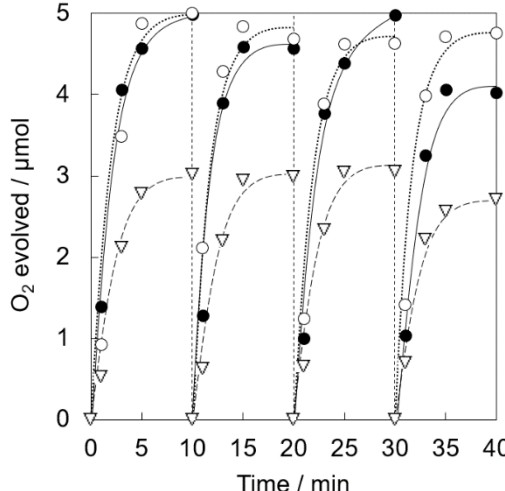

**Figure 8.** Time courses of $O_2$ evolution by visible light irradiation (white light) of a phosphate buffer solution (50 mM, 2.0 mL, pH 8.0) containing Ir(OH)$_3$/SiO$_2$NPA(20)-CA (5.0 mg, ●, solid lines), Ir(OH)$_3$/Al-SiO$_2$NPA(20)-CA (5.0 mg, ○, dotted lines) or Ir(OH)$_3$/Al$_2$O$_3$NPA(20)-CA (5.0 mg, ▽, dashed lines) in the presence of [Ru$^{II}$(bpy)$_3$]SO$_4$ (1.0 mM) and Na$_2$S$_2$O$_8$ (5.0 mM) in four repetitive experiments.

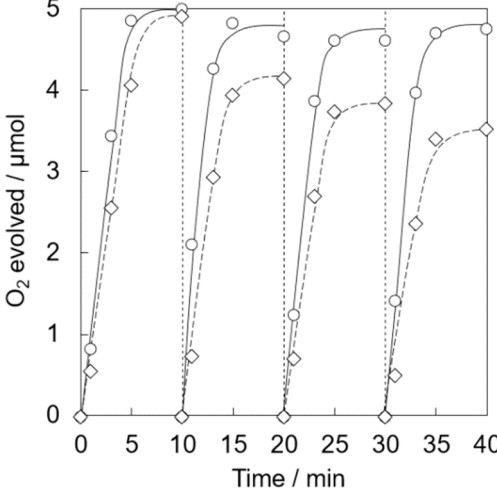

**Figure 9.** Time courses of $O_2$ evolution by visible light irradiation (white light) of a phosphate buffer solution (2.0 mL, 50 mM, pH 8.0) containing [Ru$^{II}$(bpy)$_3$]SO$_4$ (1.0 mM), Na$_2$S$_2$O$_8$ (5.0 mM) and Ir(OH)$_3$/Al-SiO$_2$NPA(20)-CA (5.0 mg, ○, solid lines) or Ir(OH)$_3$/Al-SiO$_2$NPA(10)-CA (5.0 mg, ◇, dashed lines) in four repetitive experiments.

The amount of Al$^{3+}$ used for surface modification of Ir(OH)$_3$/Al-SiO$_2$NPA(20)-CA was then optimized. Al-SiO$_2$NPA(20) modified by various amounts of Al$^{3+}$ were prepared by changing the sodium aluminate concentrations ([Al] = 1, 5, or 10 wt%) in the preparation solutions. The size of mesopores of Al-SiO$_2$NPA(20)s increased from 2.4 to 7.0 nm with increasing the concentration of sodium aluminate from 1 to 10 wt% (Table 1). Photocatalytic water oxidation was carried out under visible light irradiation of a phosphate buffer solution (50 mM, 2.0 mL, pH 8.0) containing Ir(OH)$_3$/Al-SiO$_2$NPA(20)-CA (5.0 mg, [Al] = 1, 5, or 10 wt%), Ru$^{II}$(bpy)$_3$SO$_4$ (1.0 mM), and Na$_2$S$_2$O$_8$ (5.0 mM). The $O_2$ yields at the 4th runs in the repetitive experiments with Ir(OH)$_3$/Al-SiO$_2$NPA(20)-CAs ([Al] = 1, 5, and 10 wt%) were 71%, 92%, and 88%, respectively (Figure 10). XRF measurements of Ir(OH)$_3$/Al-SiO$_2$NPA(20)-CAs showed that Ir(OH)$_3$/Al-SiO$_2$NPA(20)-CA ([Al] = 5 wt%) maintained 61% of Ir(OH)$_3$ in a fresh sample, which is higher than 36 and 53% for other Ir(OH)$_3$/Al-SiO$_2$NPA(20)-CAs

([Al] = 1 and 10 wt%), respectively (Table 1). Thus, the optimum loading amount of $Al^{3+}$ for stable immobilization of $Ir(OH)_3$ nanoparticles is around 5 wt% for $Ir(OH)_3/Al\text{-}SiO_2NPA(20)\text{-}CAs$.

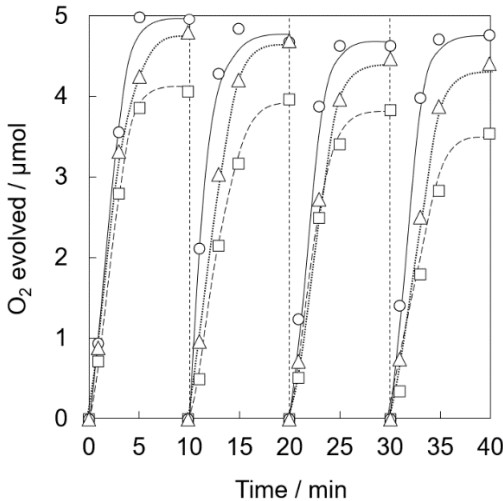

**Figure 10.** Time courses of $O_2$ evolution by visible light irradiation (white light) of a phosphate buffer solution (50 mM, 2.0 mL, pH 8.0) containing $Ir(OH)_3/Al\text{-}SiO_2NPA(20)\text{-}CA$ ([Al] = 1 wt%, 5.0 mg, □, dashed lines), $Ir(OH)_3/Al\text{-}SiO_2NPA(20)\text{-}CA$ ([Al] = 5 wt%, 5.0 mg, ○, solid lines) or $Ir(OH)_3/Al\text{-}SiO_2NPA(20)\text{-}CA$ ([Al] = 10 wt%, 5.0 mg, △, dotted lines) in the presence of $[Ru^{II}(bpy)_3]SO_4$ (1.0 mM) and $Na_2S_2O_8$ (5.0 mM) in four repetitive experiments.

## 3. Materials and Methods

### 3.1. Materials and Chemicals

All chemicals were obtained from chemical companies and used without further purification. Hydrogen hexachloroiridate(IV) hydrate, silver sulfate, sodium persulfate, sodium aluminate, disodium hydrogenphosphate, sodium dihydrogenphosphate, and sodium hydroxide were obtained from FUJIFILM Wako Pure Chemical Corporation, Osaka, Japan. LUDOX® AS-40 colloidal silica (20 nm in diameter, 40 wt% in water) and LUDOX® HS-40 colloidal silica (10 nm in diameter, 40 wt% in water), and aluminum oxide nanoparticles (<50 nm in diameter, 20 wt% in isopropanol) were purchased from Sigma-Aldrich Co. LLC, St. Louis, MO, USA. Silbol EX50 colloidal silica (50 nm in diameter, 34 wt% in water) and Silbol EX100 colloidal silica (100 nm in diameter, 25.6 wt% in water) were purchased from Fuji Chemical Co., Ltd. Toyama, Japan. Tris(2,2′-bipyridine)ruthenium(II) chloride hexahydrate was obtained from Tokyo Chemical Industry Co., Ltd., Tokyo, Japan. Tris(2,2′-bipyridine)ruthenium(II) sulfate ($[Ru^{II}(bpy)_3]SO_4$) was synthesized by adding one equivalent of silver(I) sulfate to an aqueous solution of tris(2,2′- bipyridine)ruthenium(II) chloride hexahydrate (0.13 mol, 10 mL). Ultrapure water was produced with Thermo Scientific Barnstead Smart2Pure Waltham, MA, USA, where the electronic conductance was 18.2 MΩ cm.

### 3.2. Synthesis

Synthesis of iridium hydroxide ($Ir(OH)_3$) nanoparticles: $Ir(OH)_3$ nanoparticles were synthesized according to a reported procedure with slight modification [42]. An aqueous solution of sodium hexachloroiridate(IV) ($H_2IrCl_6$, 10 mM, 25 mL) was dropped into aqueous sodium hydroxide solution (100 mM, 20 mL) using a micropump (1.0 mL/min) with vigorous stirring at 85 °C. After stirring for 1 h, insoluble aggregation appeared were removed by centrifugation. Then, methanol (100 mL) was added into the supernatant to precipitate $Ir(OH)_3$ nanoparticles. The dark-blue precipitates were collected by centrifugation and washed with ultrapure water for two times. The precipitates were dried *in vacuo* at room temperature and aged at 60 °C for 10 h.

Synthesis of mesoporous $SiO_2$ nanoparticles assembly containing $Ir(OH)_3$ nanoparticles via the co-accumulation method ($Ir(OH)_3/SiO_2NPA(20)$-CA): A dispersion of $Ir(OH)_3$ nanoparticles ([Ir] = 2.0 mM, 1.5 mL) was added to a suspension of LUDOX® AS-40 colloidal silica (40 wt%, 0.5 mL, pH 9.5) with magnetic stirring. After 30 min sonication, the dispersion was spread on a glass substrate at room temperature in a clean hood overnight. The obtained powder was washed with ultrapure water for two times and dried *in vacuo* at room temperature.

Synthesis of $SiO_2NPA(20)$ containing $Ir(OH)_3$ nanoparticles via the equilibrium adsorption method ($Ir(OH)_3/SiO_2NPA(20)$-EA): The suspension of LUDOX® AS-40 colloidal silica (40 wt%, 1.0 mL) was spread on a glass substrate at room temperature in a clean hood overnight to obtain mesoporous silica nanoparticles assembly ($SiO_2NPA(20)$) [41]. $SiO_2NPA(20)$ (200 mg) was immersed in an aqueous dispersion of $Ir(OH)_3$ nanoparticles (2.0 mL, [Ir]: 2.0 mM, pH 9.5) followed by the 30 min sonication with an ultrasound sonicator. $SiO_2NPA(20)$ was collected by centrifugation and washed with ultrapure water for two times. The wet powder was dried *in vacuo* at room temperature.

Synthesis of silica-alumina ($Al-SiO_2$) nanoparticles: $Al-SiO_2$ nanoparticles were prepared according to a reported procedure with slight modification [41]. A dispersion of LUDOX® AS-40 colloidal silica (40 wt%, 1.93 mL) was slowly added to an aqueous solution of sodium aluminate (13 mM, 9.6 mL, 48 mL, or 96 mL for [Al] = 1 wt%, 5 wt%, or 10 wt% samples, respectively) at room temperature. The resulting dispersion was magnetically stirred for 24 h and used for the next procedure without further purification.

### 3.3. Characterization

The atomic ratio of iridium and silicon in each catalyst was determined using a Shimadzu EDX–730 X-ray fluorescence spectrometer. TEM images and elemental mapping images of catalysts were obtained using a JEOL JEM–2100 equipped with a field-emission gun with an accelerating voltage of 200 kV with an X-ray energy dispersive spectrometer. Thin pieces of catalyst were fixed on a Cu-mesh microgrid coated with an amorphous carbon supporting film. Nitrogen ($N_2$) adsorption–desorption isotherms were measured at 77 K using a MicrotracBEL Belsorp-mini II within a relative pressure range from 0.01 to 101.3 kPa. The mass of a sample was as much as ca. 15 mg for adsorption analyses after pretreatment at 150 °C for 1 h. The sample was exposed to a mixed gas of helium and $N_2$ with a programmed ratio and adsorbed amount of nitrogen was calculated from the change of pressure in a cell after reaching the equilibrium. DLS experiments were conducted at room temperature using a Malvern Zetasizer Nano S90.

### 3.4. Photocatalytic Water Oxidation

A typical procedure for photocatalytic water oxidation is as follows. A phosphate buffer solution (50 mM, 2.0 mL, pH 8.0) containing $Ir(OH)_3/SiO_2NPA(20)$-CA (5.0 mg, [Ir] = 0.072 mM), sodium persulfate ($Na_2S_2O_8$, 5.0 mM), and tris(2,2′-bipyridine)ruthenium(II) sulfate ([$Ru^{II}(bpy)_3$]$SO_4$, 1.0 mM) was flushed with Ar for 15 min in dark. The buffer solution was photoirradiated for a certain time with a RelyOn white LED light (130 mW, $\lambda$ > 400 nm) positioned perpendicular to a cuvette. The distance between the lamp and sample cell was 3.0 cm, thus the light intensity was 33.2 mW cm$^{-2}$. The gas in a headspace was analyzed by using a Shimadzu GC-2014 gas chromatograph with a thermal conductivity detector to determine the amount of $O_2$ evolved. Recycling performance was evaluated by adding a phosphate buffer solution (50 mM, 2.0 mL, pH 8.0) containing $Na_2S_2O_8$ (5.0 mM) and [$Ru^{II}(bpy)_3$]$SO_4$ (1.0 mM) to a catalyst taken out from the reaction solution by centrifugation.

## 4. Conclusions

Iridium hydroxide ($Ir(OH)_3$) nanoparticles, which act as an active catalyst for photocatalytic water oxidation in the presence of tris(2,2′-bipyridine)ruthenium(II) ion and persulfate ion as a photosensitizer and a sacrificial electron acceptor, respectively, were immobilized in mesopores of a $SiO_2$ nanoparticles assembly ($SiO_2NPA$). The co-accumulation method, in which $Ir(OH)_3$ nanoparticles were immobilized

in the mesospaces during the formation of $SiO_2NPA$, is a promising way to fabricate the stable catalyst compared with the conventional equilibrium adsorption method. Further enhancement of catalytic stability was observed by tuning the size of $SiO_2$ nanoparticles and the surface alumination of $SiO_2NPA$. We propose here a new approach to enhance stability of catalytic nanoparticles by using spherical nanoparticles assemblies as porous support.

**Supplementary Materials:** The following are available online at http://www.mdpi.com/2073-4344/10/9/1015/s1, Figure S1: Particles size distribution, Figure S2: Zeta potential, Figure S3: Nitrogen adsorption–desorption isotherms.

**Author Contributions:** Conceptualization, Y.Y.; methodology, H.T. and Y.Y.; formal analysis, G.S. and H.T.; investigation, G.S. and H.T.; writing—original draft preparation, G.S. and H.T.; writing—review and editing, H.T. and Y.Y.; project administration, Y.Y.; funding acquisition, Y.Y. and H.T. All authors have read and agreed to the published version of the manuscript.

**Funding:** This work was supported by Innovative Science and Technology Initiative for Security, ATLA, Japan (grant numbers J161000157 and J191047001 (to Y.Y.)); JSPS KAKENHI (grant numbers JP16H02268 (to Y.Y.), JP19KK0144 (to Y.Y. and H.T.), JP19K15591, and JP20H05110 (to H.T.)); and ENEOS hydrogen trust fund (to Y.Y.).

**Acknowledgments:** We thank Masatsugu Ishimoto from Osaka City University for his support throughout the TEM measurements.

**Conflicts of Interest:** The authors declare no conflict of interest.

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
