# Peer review of "Immobilization of Ir(OH)3 Nanoparticles in Mesospaces of Al-SiO2 Nanoparticles Assembly to Enhance Stability for Photocatalytic Water Oxidation"

_catalysts, doi:10.3390/catal10091015_

Round 1
Reviewer 1 Report
This paper presents a study of Ir(OH)3/SiO2NPA assemblies in the application of photocatalytic water oxidation. Ir(OH)3/Al-SiO2NPA has been proven to effectively suppress the leaching of Ir(OH)3 and reasonable enhancement of stability during photocatalytic water oxidation was obtained for the silica or silica-alumina nanoparticles. The author also conducted a series of experiments to verify the enhancement of stability. There are however several aspects in the manuscript that need further clarification before it is suitable for publication in Catalysts.
Following are the comments listed to be considered by the authors:
- My main concern about this paper comes from the arrangement of the manuscript. Some of the data presented in the Supplementary Material is critical in explaining the change in particle size and mesoporous structures. The authors should consider transferring some contents to the manuscript instead of citing the Supplementary Material numerous times.
- Page 4, Line 125 & Page 9, Line 289: What’s the light source and intensity of the white-light radiation? More specific information on the photocatalytic water oxidation experiment is preferred here.
- Page 5, Line 148: The caption of Figure 4S is confusing. Is it just Ir(OH)3/SiO2NPA(20)-CA dispersed in water stirred at 400 rpm for 10 min, 20 min, 30 min, and 40 min without reaction or reaction solution obtained at 10 min, 20 min, 30 min, and 40 min reaction time?
- Page 5, Line 150: In Figure 4S, there is only one peak in the 10 min (solid line) and 20 min (dashed-dotted line) curve and they are highly overlapping. This is weird because there is no transition stage showing the change of particle size. Does this mean the change in particle size only takes place at 20-30 min? The authors should add an explanation.
- Page 6, Line 164: “Transmission electron microscope (TEM)” Abbreviations should be defined once.
- Figure 5, 6, and 7, can the authors use different colors or hollow/solid plots to distinguish the data points?
Reviewer 2 Report
Most of all parts described, well. But it requires HRTEM and DES mapping images, due to TEM images are clear. And, it needs english revision by native or native company.
